# Patients' experiences of a Virtual Fracture Assessment Clinic Pathway: A qualitative study

Darragh Carolan[1]*, Shane O'Neill[2], Breon White[1], Frank Lyons[2], Grainne Colgan[2], Vinny Ramiah[3], Dervilla Danaher[1], Gillian Yeowell[4]

1 Physiotherapy Department, Mater Misericordiae University Hospital, Dublin, Ireland, 2 Department of Orthopaedic Surgery, Mater Misericordiae University Hospital, Dublin, Ireland, 3 Emergency Department, Mater Misericordiae University Hospital, Dublin, Ireland, 4 Department of Health Professions, Faculty of Health and Education, Manchester Metropolitan University, Manchester, United Kingdom

* darraghcarolan@mater.ie

## Abstract

### Background and objectives

Existing literature has demonstrated the efficacy of a virtual fracture assessment clinic pathway, however there is limited research exploring patients' experiences of a complete pathway, from initial presentation at the Emergency Department or Injury Unit to discharge. The aim of this study was to explore the experiences of patients who have recently sustained a stable peripheral limb fracture, having received care across a complete virtual fracture assessment clinic pathway, in order to improve patient care.

### Methods

One-to-one semi-structured interviews were completed via recorded phone and video calls, with a purposive sample of 12 participants. Interviews were completed until data saturation was achieved. Interviews were transcribed verbatim and data was analysed using thematic analysis.

### Results

Six overarching themes were identified; 'trust' (in the pathway and management plan), 'conflicting advice' (on diagnosis and management plan), 'information' (need for more basic information), 'severity of injury' (participants' perceptions of the severity of their injuries), 'reassurance' (through follow-up x-rays and physiotherapy consultations) and 'efficiency'.

### Conclusions

This is the first qualitative study exploring patients' experiences of a complete virtual fracture assessment clinic pathway. Patients' experiences may be improved through patient education on the pathway process and providing standardised injury-specific patient information documentation. Regular communication between different healthcare

**Data availability statement:** All relevant data are within the manuscript and supplementary information files. Relevant anonymised quotes from participants have been included in the results section of the manuscript. As per ethical approval obtained from the Mater Misericordiae University Hospital Institutional Review Board (reference number: 1/378/2387) and Manchester Metropolitan University Faculty Ethics Committee (reference number:

58689), anonymised versions of interview transcripts are saved on the Mater Misericordiae University Hospital computer system. Full anonymised transcripts can be made available on request. Requests can be sent to the Hannah King, Administrator, Institutional Review Board, Mater Misericordiae University Hospital, Eccles Street, Dublin 7, D07 R2WY, Ireland, hannah-king@mater.ie, +353 18032971.

**Funding:** The lead author (DC) has received funding from the Mater Misericordiae University Hospital https://www.mater.ie/ and the Musculoskeletal Association of Chartered Physiotherapists (MACP) UK https://www.macpweb.org/research/awards/17/2024-msc-pgd-portfolio-route-bursaries towards their academic fees for the Masters degree in which this study was completed as part of a dissertation submission. No funding was used to complete this study. The funders had no role in study design, data collection and analysis, decision to publish or preparation of the manuscript.

**Competing interests:** The authors have declared that no competing interests exist.

professionals involved in the pathway may reduce conflicting advice. Developing an opt-in physiotherapy service and providing patients with a standardised text message informing patients when their virtual consultation will occur may also improve patients' experience of the pathway. Establishing a referral pathway with a fracture liaison service is also recommended to further enhance a virtual fracture assessment clinic pathway. Future research is needed to investigate how therapeutic relationships can be developed when care is delivered virtually and to explore the experiences of patient cohorts who were not included in this study.

## Introduction

### Background

One of the first virtual fracture assessment clinic (vFAC) pathways described in the literature was established in Glasgow in 2011 as an alternative pathway for managing stable fractures [1]. In this original pathway, patients presenting to the emergency department (ED) were treated with a splint or sling, educated on their injury and were discharged or referred to the vFAC [1]. In the vFAC, an orthopaedic consultant reviews patient's imaging and determines whether the patient should be discharged, referred to a nurse-led clinic or a specialist consultant-led clinic. vFAC teams have evolved to include extended practice physiotherapists [2], which means patients can be discharged at a vFAC consultation directly to an in-person physiotherapy consultation.

vFAC pathways can reduce the need for in-person fracture clinic appointments [3] and unnecessary follow-up x-rays [4]. Despite the clear efficacy of the vFAC pathway [5], there is a dearth of qualitative research exploring patients' experiences of the pathway. Without an understanding of patients experiences of a vFAC pathway, it is difficult to determine how to optimise this experience for future patients.

### Knowledge from existing research

There is a dearth of literature that has explored patients' experiences of a vFAC pathway using in-depth qualitative methods. Willinge et al. [6], utilised semi-structured interviews and thematic analysis to explore patients' experiences of the ED and vFAC consultations. They found that participants were satisfied with waiting time in ED and the time interval between ED presentation and the vFAC consultation, without the need to attend the hospital for this consultation. Information received was adequate, but participants requested further advice on self-management of their recovery. Participants were confident in healthcare workers abilities; however conflicting advice was problematic. The main limitation of the study was that only experiences of the ED and vFAC consultations were investigated. Exploring the remaining phases of the pathway, which may include direct discharge after the vFAC consultation, in-person fracture clinic or physiotherapy consultations and the recovery process, would significantly improve the transferability of these findings.

Other studies exploring patients' perspectives of a vFAC pathway are limited to injury-specific patient reported outcome measures (PROMs) [7–12] and patient satisfaction questionnaires [7–23]. Questionnaire based studies have established satisfaction with care received on a vFAC pathway between 75%-97.5% [8,18]. Patient satisfaction with information received regarding their injury and management varies between 86.4%-98% [7,11]. In contrast, another questionnaire-based study [23] highlighted 40% of patients desired further information. Despite the convenience of a vFAC, 17–28% of patients report a preference for attending an

in-person fracture clinic [13,24]. A retrospective study [25] investigated patients requiring in-person fracture clinic follow-up after unsuccessful discharge at the vFAC consultation, 98.11% were managed with reassurance and physiotherapy. Further research is needed to investigate how reassurance can be provided as part of the pathway, to reduce the need for unnecessary in-person fracture clinic consultations. Insights gained from patient satisfaction questionnaires and PROMs are limited to Likert scales or questions with yes/no answers. These formats provide limited insight into the patient experience and only explore concepts predetermined by the researcher.

### Gaps in existing literature

Existing literature focuses on patients' experiences of the ED and vFAC consultations. There is a need for further qualitative research to explore patients' experiences of a complete pathway, in order to improve this pathway for future patients. It is important to explore why some patients report a preference for in-person fracture clinic consultations, this may help to improve the vFAC pathway and reduce patients requesting unnecessary in-person fracture clinic consultations. Furthermore, many patients may not be referred for an in-person physiotherapy consultation. Given that physiotherapists have been described as the main source of information for patients when recovering from injury [26], exploring what information patients require during the pathway may also improve patient experiences and outcomes.

### Study aim

The aim of this study was to explore the experiences of patients' who have recently sustained a stable peripheral limb fracture, once they have received care across a complete virtual fracture assessment clinic pathway, in order to improve patient care.

## Methods

The Standards for Reporting Qualitative Research (SRQR) framework [27] was utilised to ensure the study methodology is transparent, reliable and repeatable for others [28].

### Design

A qualitative design using an interpretivist paradigm was employed to explore patients' experiences of a complete vFAC pathway. Interpretivism seeks to understand how participants subjectively interpret reality and create meaning from their own perspectives [29], and as such, was appropriate to address the research aim.

The lead author (DC) who caried out all interviews and thematic analysis is a clinical specialist physiotherapist working in the vFAC pathway. This provided them with insider knowledge into common concerns expressed by patients regarding the pathway.

To mitigate against the possible influence of a researcher-participant power imbalance on open communication [30], the lead author (DC) did not recruit patients whose care they were involved in. Correspondingly the lead author (DC) did not disclose to participants that they are a physiotherapist to avoid the negative influence this may have had on participants openly reporting their experiences of physiotherapy [31]. The lead author (DC) completed field notes after interviews, to reflect on the influence of their reflexivity [32].

This was a single site study recruiting participants from Mater Misericordiae University Hospital in Dublin, Ireland. Ethical approval was granted by the Mater Misericordiae University Hospital Institutional Review Board (reference number: 1/378/2387) and Manchester Metropolitan University Faculty Ethics Committee (reference number: 58686).

### Inclusivity in global research

Additional information regarding the ethical, cultural and scientific considerations specific to inclusivity in global is included in the supporting information (S1 Checklist).

### Participant recruitment

Participant recruitment was undertaken between 19/12/2023-09/05/2024. To ensure the sample was reflective of patients treated via the hospital's vFAC pathway, a purposive sample was recruited in regard to the demographic details and management plans for patients referred to the pathway in 2023 (S2 Table). The sample was recruited across three different orthopaedic consultant-led (SO'N, FL, GC) vFACs to mitigate the influence of a single consultant on participants' experiences. Sample size was determined by data saturation, the point where no additional themes are obtained from further interviews [33] and no new themes were identified from interviews of two participants in a row [34].

Written informed consent was obtained in line with good clinical practice guidelines [35], authors (DC, GY) completed good clinical practice training in advance of commencing the study. According to the 2015 Assisted Decision-Making (Capacity) Act in Ireland, capacity is assumed unless it has been proven otherwise [36].

Inclusion criteria:

- Participants who had recently sustained a stable peripheral limb fracture and had received care across a complete vFAC pathway

- Aged 18 or above

- Any sex was included

Exclusion criteria:

- Participants who could not communicate in English

- Participants referred to the vFAC from a different hospital or Injury Unit (IU) (as they may have had different experiences in the ED/IU at another site)

- Participants with cognitive impairments would not have been able to provide informed consent

A vFAC physiotherapist (BW) approached potential participants about participating in the study. The lead author (DC), who conducted all interviews, did not approach patients whose care they were involved in as this may have led to a risk of coercion [37]. If a potential participant was agreeable to discussing participation, they were contacted by the lead author (DC) by phone to discuss the study further. Potential participants were posted a participant information leaflet (PIL), consent form and a stamped addressed envelope to return the completed consent form. A follow-up phone call was completed by the lead author (DC) one week later to allow potential participants time to consider participation [37].

### Data collection methods

One-to-one interviews were completed with participants to facilitate a deep and rich understanding of their experiences [38]. Semi-structured interviews were conducted over recorded Zoom video calls or telephone calls. Video calls were recorded using the Zoom recording facility; telephone calls were recorded by placing the telephone on speaker phone and recording audio over Zoom.

An interview guide (S3 Table) was developed from a review of literature and the expertise of the research team. The interview guide was then modified following patient and public

involvement (PPI). This included patients who had recently been managed via the vFAC pathway and from other members of the vFAC team. Two pilot interviews were undertaken to test the interview guide and for the lead author (DC) to gain interview experience [39]. As only minor changes to the interview guide were required, to remove repetitive questions and improve the sequence of questions, the pilot interview was included in the study. As the study progressed, the findings from one interview were iteratively fed into subsequent interviews to enable the exploration of unanticipated findings.

As participants had a traumatic injury and may become distressed during interviews, they may need to be safeguarded if concerns are identified [40]. A distress protocol was created in the event that any psychological distress was identified. Before interviews commenced participants were reminded of the distress protocol and advised they could decline answering any questions. If participants became distressed or there were any concerns for their psychological safety, the interview would be paused to allow participants to take a break and the interview could be ceased if needed.

### Data analysis

Thematic analysis using Braun and Clarke's six phase framework was chosen as the data analysis process as it allows for flexible and transparent analysis and is a straightforward approach for inexperienced researchers to learn [41]. Data was analysed with the assistance of NVivo software (version 14.23.4).

Participants were provided with a copy of their anonymised interview transcript and the overall analysis of themes. This member checking process enhanced dependability by allowing participants an opportunity to feedback on the accuracy of the authors interpretations [42]. Direct quotations are provided to enhance the credibility of the authors interpretations [43]. Negative cases, where experiences of some participants differed from the majority of participants, are reported to allow for authenticity of a spread of reported experiences [44].

## Results

### Participants

In total 26 potential participants were approached; a summary of the recruitment process is provided in S4 File. Saturation was reached by the 10th interview, when no new themes were identified [33]. Two further participants were interviewed as participants had already been recruited and this enabled the lead author (DC) to confirm saturation. Twelve participants were interviewed, seven were female, six had upper limb fractures. Interviews were between 30–75 minutes in duration, with a mean time of approximately 60 minutes. Demographic details of participants can be found in Table 1. Participants are named using pseudonyms to ensure quotes feel personal and to allow the reader to follow the interview narrative [45].

### Overarching themes

Across the data set, six overarching themes were identified: 'trust', 'conflicting advice', 'information', 'reassurance', 'severity of injury' and 'efficiency'. A thematic map (Fig 1) identifies the relationship between themes and subthemes. Each theme is presented with pseudonymised participant quotes to support the findings.

### Trust

Trust was a central theme linked to participants' experiences. Despite the majority of participants not attending an in-person fracture clinic, many still felt they received appropriate care and trusted the virtual process.

**Table 1. Participant demographic details.**

| Pseud-onym | Sex | Age Range | Occupation | Geographi-cal Location | Number of Days From ED/IU Presentation to Interview Date | Injury | Location of Initial Presentation | Orthopaedic Consultant Speciality | vFAC Outcome | Interview Method |
|---|---|---|---|---|---|---|---|---|---|---|
| Rita | Female | 65+ | Office Based | Urban | 53 | Distal Radius Fracture | IU | General | In-Person Fracture Clinic and Physiotherapy | Phone |
| Lily | Female | 65+ | Retired | City | 40 | Clavicle Fracture | IU | General | Physiotherapy | Zoom |
| Catherine | Female | 65+ | Retired | Urban | 59 | Distal Radius Fracture | IU | Upper Limb | In-Person Fracture Clinic and Physiotherapy | Phone |
| Alice | Female | 25-34 | General Operative | Urban | 50 | 5th Metatarsal Fracture | IU | Lower Limb | Physiotherapy | Phone |
| Emma | Female | 55-64 | Education | Urban | 31 | Wrist Osteoarthritis | IU | Upper Limb | Physiotherapy | Zoom |
| Anna | Female | 25-34 | Retail | City | 50 | Talus Fracture | IU | General | Physiotherapy | Phone |
| Dennis | Male | 65+ | Retired | Urban | 24 | Radial Head Fracture and 5th Metatarsal Fracture | ED | General | Discharged | Phone |
| Alex | Male | 25-34 | Tradesper-son | Urban | 40 | Great Toe Fracture | IU | General | Discharged | Phone |
| Paul | Male | 18-24 | Student | Urban | 65 | Great Toe Fracture | IU | Lower Limb | Discharged | Zoom |
| Barry | Male | 45-54 | Office Based | Urban | 20 | Great Toe Fracture | IU | General | Discharged | Zoom |
| Sophie | Female | 25-34 | Healthcare | Urban | 81 | 5th Metatarsal Fracture | IU | Lower Limb | In-Person Fracture Clinic | Zoom |
| John | Male | 18-24 | Student | City | 142 | Radial Head Fracture and Scaphoid Fracture | IU | General | In-Person Fracture Clinic and Physiotherapy | Phone |

ED = emergency department, IU = injury unit, vFAC = virtual fracture assessment clinic.

> I'm all for it…I would trust doctors…to not do it for the sake of saving time and money… if there is any doubt they are going to- 'come in let me have a look at it'. Barry (Great toe fracture, discharged with no further follow-up).

In contrast, participants reporting a negative experience cited a lack of trust in their care.

> I just felt like [they] didn't believe me…I think [they were] just unsure about what to actually do with me. Anna (Talus fracture, discharged to physiotherapy).

Trust was a recurring theme for participants when they received conflicting diagnoses from different healthcare professionals.

> I wasn't sure who to trust or who to believe…I'm still not one hundred percent convinced what I have. Emma (Referred to vFAC as a scaphoid fracture but diagnosed as wrist osteoarthritis by the orthopaedic consultant, referred to physiotherapy)

## Conflicting advice

Conflicting advice from difference healthcare professionals was a source of anxiety. One participant was discharged from the IU diagnosed with an ankle sprain but was later informed a radiology report confirmed a fracture.

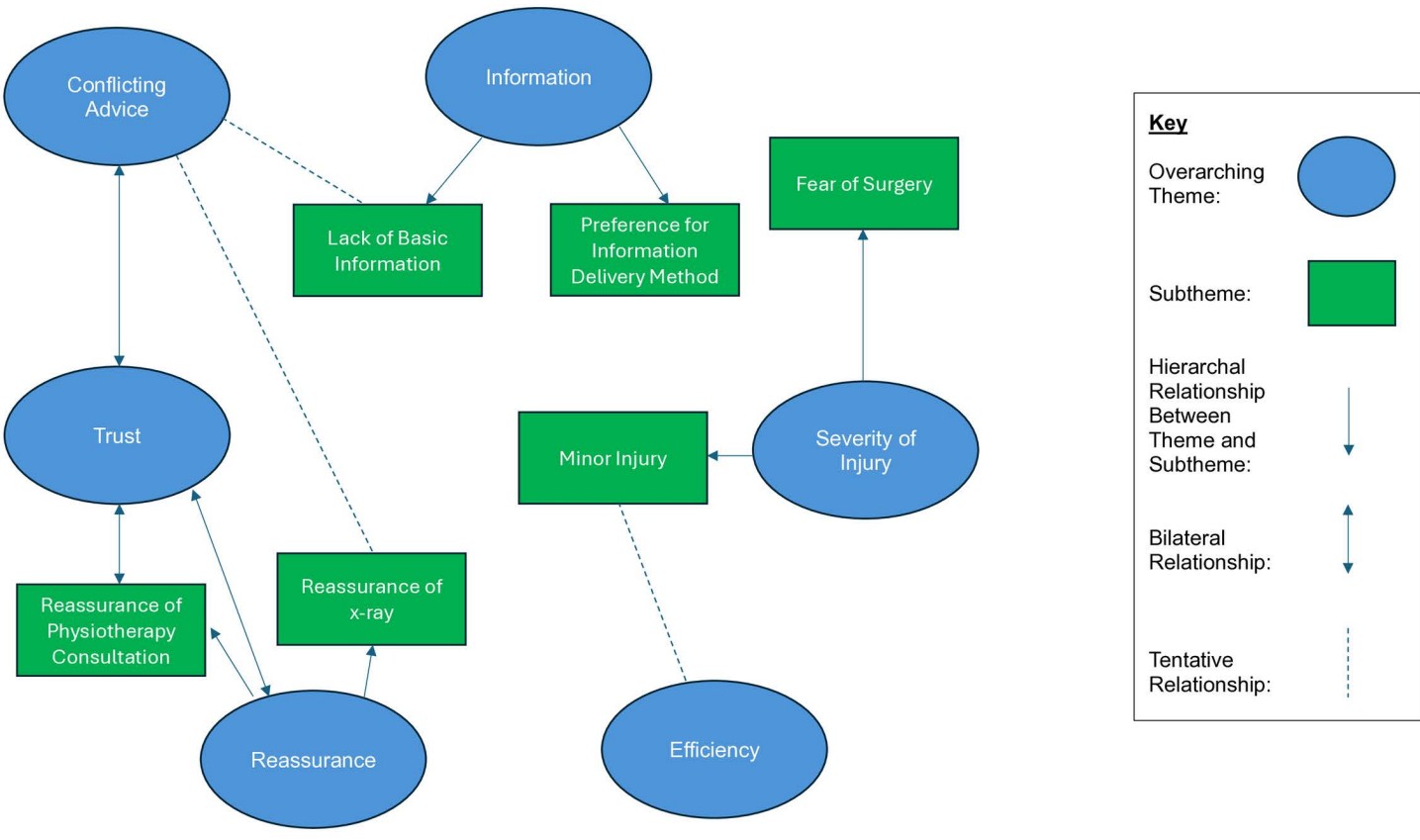

**Fig 1. Thematic map.**

She did ring me two days later she did apologise and said 'oh no the [radiologist] actually says there's a break there that we missed'. Sophie (5th metatarsal, referred to in-person fracture clinic)

Some participants received conflicting advice on their management plan, particularly how long to wear immobilisation devices for.

[the vFAC] was like use the boot or the [surgical shoe] for four weeks…I got told in the [IU] after four days you'll probably take the thing off…and I was like 'ok well I didn't really know that' cause at that point I stopped using it. Paul (Great toe fracture, discharged with no further follow-up)

Despite several reports of conflicting advice, other participants reported receiving consistent advice.

Everything was what they were saying in A&E over the [vFAC consultation] as well. Dennis (Radial head and 5th metatarsal fracture, discharged without further follow-up)

## Information

Participants expressed a need to receive more information about their injury.

I didn't feel there was much communication going on, it was just 'yeh you have a fracture, here you go here's your page the [vFAC] will follow up with you'. Emma (Referred to vFAC

as a scaphoid fracture but diagnosed as wrist osteoarthritis by the orthopaedic consultant, referred to physiotherapy)

Participants highlighted a need for basic information on immobilisation devices.

> They could have actually given me more like advice on what to…with the sling…I suppose that would be a good place just assume I know nothing. John (Radial head and scaphoid fractures, referred to in-person fracture clinic and physiotherapy)

Some participants expressed a desire to receive information electronically.

> An email would be handy…just to keep it digitalised because… letters get lost. Paul (Great toe fracture, discharged with no further follow-up)

While others preferred the simplicity of receiving written information.

> I was quite happy with it on a page, I'm of that era ((laughs)). Rita (Distal radius fracture, referred to in-person fracture clinic and physiotherapy)

### Reassurance

Reassurance was important for participants to alleviate fears of poor recovery. This reassurance often came from objective assessments such as x-rays and in-person physiotherapy consultations. Participants referred to the in-person fracture clinic were reassured to see evidence of healing on their follow-up x-rays.

> They showed me…the x-ray…I was happy to get it done, I like to see how it's knitting and how it's doing. Catherine (Distal radius fracture, referred to in-person fracture clinic and physiotherapy)

Conversely for participants not referred to an in-person fracture clinic, the lack of a follow-up x-ray was a source of anxiety.

> I did ask was there going to be a follow-up x-ray to make sure it's healing alright…but for a less minor break they said 'no'. Lily (Clavicle fracture, referred to physiotherapy).

An in-person physiotherapy consultation was another source of reassurance.

> I was quite nervous even walking on it before I met [the physiotherapist] and [the physiotherapist] was like 'it's fine'…even [the physiotherapist] saying that I was like 'oh good'. Anna (Talus fracture, referred to physiotherapy).

Participants who were referred for an in-person physiotherapy consultation were asked if they would rather the option of self-managing with a home exercise programme, many stated a preference for an in-person physiotherapy consultation.

> I'd probably go with the [in-person] physio just for me just for reassurance in me own head just that I'm walking and doing everything as correctly you know. Alice (5th metatarsal fracture, referred to physiotherapy)

## Severity of injury

Participants' perceptions of the severity of their injury were an important factor in their experience of virtual care. Many participants felt their injuries were relatively minor, which may have reduced the desire for an in-person fracture clinic consultation.

> I was happy with the phone call sure there's no need for me to end up in the hospital if I don't have to… there's other patients worse than I am. Dennis (Radial head and 5th metatarsal fractures, discharged with no further follow-up)

Despite this perception of a minor injury, other participants were concerned that they may require surgery.

> Because I was told [the referral] was going onto orthopaedics- surgical orthopaedics I didn't know whether I needed an operation. Sophie (5th metatarsal fracture, referred to in-person fracture clinic)

## Efficiency

Efficiency was a central theme throughout the pathway, particularly in participants' experiences of the IU.

> It's exactly what you want if you're only going in with- it could be a broken wrist or broken toes you want it quick and fast in and out like. Alex (Great toe fracture, discharged without further follow-up)

Many participants were pleased with the efficiency of the vFAC phone call, instead of attending an in-person fracture clinic.

> I didn't have to take time off. I didn't have to reschedule anything. I didn't have to sit for hours in a clinic. Emma (Referred to vFAC as a scaphoid fracture but diagnosed as wrist osteoarthritis by the orthopaedic consultant, referred to physiotherapy).

Despite the convenience of virtual follow-up, some participants stated a preference for an in-person fracture clinic consultation.

> I'd prefer to come in…I think you'd find that a lot with older people that they prefer to be seeing the doctor. Catherine (Distal radius fracture, referred to fracture clinic and physiotherapy).

Despite the convenience of a vFAC consultation several participants found it difficult not knowing when this consultation would occur.

> I actually missed his call twice. I called him back and I missed his call. I called him back, then we got each other the third time. Barry (Great toe fracture, discharged with no further follow up)

Participants suggested a text message confirming when the vFAC consultation would occur, would inform them when to expect the vFAC phone call and allay any fears of a scam call, which was another concern for participants.

At least by getting a text message… you know it's not going to be a scam that way and you know to answer your phone around that time from a private number. Sophie (5th metatarsal fracture, referred to in-person fracture clinic)

## Discussion

This study explored the experiences of patients' who had recently sustained a stable peripheral limb fracture, once they have received care across a complete vFAC pathway, in order to improve patient care. Altogether six overarching themes were identified; 'trust', 'conflicting advice', 'information', 'reassurance', 'severity of injury' and 'efficiency'.

Trust was a central theme in this current study, participants reported a positive experience when they trusted they were receiving appropriate care. A similar subtheme was identified in another qualitative study [6] where participants reported being confident in the care they received from healthcare workers during their ED and vFAC consultations. Trust is a common finding across qualitative research in patients' experiences in fracture wards and in-person fracture clinics [46, 47]. Conversely, participants in this current study reporting a negative experience cited a lack of trust in the pathway. Developing a therapeutic relationship with patients when care is delivered virtually may be challenging and this may be a factor as to why some participants in this current study expressed a preference for an in-person fracture clinic consultation. Similarly, a study exploring a telephone physiotherapy service, participants reported difficulty developing a therapeutic relationship with a physiotherapist over the phone [48]. Understanding how a therapeutic relationship can be fostered when care is delivered virtually may help to improve patient outcomes.

Trust in this current study was particularly challenged when participants received conflicting advice on their diagnosis and management plan. Correspondingly, a study investigating outcomes and satisfaction levels for patients with hand injuries, managed via a vFAC, noted some patients were initially diagnosed as having a fracture in the ED/IU but were subsequently advised they did not have a fracture by the vFAC [9]. This highlighted the need to educate patients on the vFAC process, to provide patients with a clear message, particularly when there is a possibility of conflicting diagnoses [9]. Conflicting advice on management plans has also been reported in research exploring patients' experiences post-major trauma [46].

Participants in this current study expressed a desire for more basic information about their injury and managing immobilisation devices. Satisfaction with information received from vFACs varies in existing literature, a quantitative study exploring patients' satisfaction with a UK vFAC reported 40% of patients wanted more information about their fracture and management plan [23], whereas a similar study in an Irish vFAC reported satisfaction with information received at 98% [11]. A vFAC service in the UK have developed a website to provide patients with additional information [9], delivering information digitally may be beneficial as 42% of participants in a vFAC service in Ireland did not recall receiving a written patient information leaflet in the ED [13]. As highlighted by participants in this current study, it is imperative to also provide information in paper form, as this may be the preferable method of information provision for some patients. Providing patients with more information about their injury and using immobilisation devices may help to improve their experience of a vFAC pathway and lead to better outcomes.

Several participants in this current study felt a follow-up x-ray would provide reassurance, which may have led to their desire for in-person fracture clinic follow-up. This highlights the need to educate patients that many stable fractures managed through a vFAC pathway do not require follow-up x-rays [4]. In a study exploring patient's satisfaction with self-management

of a distal radius fracture through a UK vFAC [16], several participants contacted the vFAC enquiring if follow-up x-rays were required, this service has now updated their patient information leaflets to advise follow-up x-rays are not required. While some participants in this current study were satisfied to be discharged without a physiotherapy consultation, many participants expressed a desire to see a physiotherapist for reassurance. Physiotherapists have previously been described as the main source of advice for patients recovering post-traumatic injuries, participants not referred to physiotherapy were more apprehensive about their recovery [26]. Therefore, the option of an opt-in physiotherapy service may be beneficial for patients who require further reassurance.

The need for reassurance in this current study was influenced by participants' perceptions of the severity of their injuries, several participants were concerned they may require surgery. Data from the vFAC service in this current study indicates only 1% of patients referred to the pathway in 2023 required surgery (S2 Table). In a retrospective study investigated patients directly discharged by a UK vFAC who re-presented to an in-person fracture clinic for further follow up, only 1.8% of these patients required surgery [25]. It is important to reassure patients that surgery is rarely required on a vFAC pathway. The majority of participants in the current study felt their fracture was relatively minor which may have reduced the desire for an in-person fracture clinic consultation. Comparably, some patients express a preference for a virtual general practitioner consultation over an in-person consultation when they feel their medical complaint is relatively minor [49]. Previous qualitative studies have highlighted the need for psychological support for participants post-fracture [50], while other participants have reported feeling dismissed when managed with low levels of immobilisation such as slings as opposed to plaster cast [51]. These experiences were not reflected in this current study, possibly due to participants' perceptions of a less severe injury, that does not require the same level of care.

Efficiency was cited by participants in this current study as a positive experience of the vFAC pathway, as opposed to attending an in-person fracture clinic. This finding was confirmed in a similar study where participants were relieved they did not have to attend an in-person fracture clinic, particularly with reduced mobility post-fracture [6]. A questionnaire investigating satisfaction with an Irish vFAC reported 64% of patients (or parents of patients) would have needed time off work to attend an in-person fracture clinic [11]. Additionally, in the current study participants highlighted that they were unaware when the vFAC consultation would occur. A similar concern has been highlighted by patients awaiting a virtual consultation with a general surgeon where virtual calls often came at inopportune times where patients were caught off guard and not prepared to ask questions [52]. Therefore, providing a standardised text message informing patients when their virtual consultation will occur may enhance patients' experience and satisfaction with the vFAC pathway.

One pertinent concern raised by a single participant in the current study was the lack of investigation regarding the possibility of sustaining a fragility fracture. Previous research has suggested the need for a fracture liaison nurse specialist as part of a vFAC service [53]. Therefore, this concern that fragility fractures are potentially being undiagnosed in a vFAC service that do not have a referral pathway to a fracture liaison service is worthy of further investigation. Including information on fragility fractures in standardised patient information leaflets may encourage patients to discuss bone health screening with their general practitioner.

The findings of this study highlight that patients may benefit from education on the vFAC pathway process, including information around the possibility of receiving conflicting diagnoses and the fact that follow-up x-rays and surgical management are rarely required on a vFAC pathway. Regular communication between different healthcare professionals involved in the vFAC pathway may help to improve consistency of advice. Developing standardised

injury-specific patient information documentation, including advice on using immobilisation devices may be helpful for patients managed on a virtual pathway. Providing this information via a website may also complement paper documentation. Finally, efficiency of care may be improved through the use of simple technology such as sending a standardised text message to patients, advising when the vFAC consultation will occur.

Future research is needed to investigate how therapeutic relationships can be developed when care is delivered virtually. Moreover, future research should explore the experiences of patient cohorts who were not included in this study such as patients under the age of 18, patients with cognitive impairments and patients from marginalised communities such as foreign nationals who do not speak English, asylum seekers, homeless patients, illicit drug users and prisoners.

A strength of this study is that through a purposive sample, we successfully recruited participants similar in profile to patients referred to the hospital's vFAC service in 2023. Furthermore, the sample was reflective of the existing literature from Ireland and the UK [21,53], thus increasing the transferability of the findings of this study. However, there were some weaknesses to this study including the necessity for participants to provide informed consent excluded patients with cognitive impairments and patients under the age of 18, reducing the transferability of the study findings to these cohorts. The confines of ethical approval dictated it was not possible to recruit vulnerable patients from marginalised communities such as foreign nationals with low levels of English, asylum seekers, homeless patients, illicit drug users and prisoners also reducing the transferability of the study findings, particularly as these populations are frequently referred to the vFAC pathway at the study site. Additionally, several potential participants approached about participation did not undertake the study, which may have led to a participation bias [54], as the experiences of patients who did not complete interviews may have led to the development of alternative themes.

## Conclusion

To the authors' knowledge this is the first qualitative study exploring patients' experiences of a complete vFAC pathway. In total six overarching themes were identified; trust, conflicting advice, information, severity of injury, reassurance and efficiency. Patients' experiences of a vFAC pathway may be improved through patient education on the vFAC pathway process, regular communication between different healthcare professionals involved in the vFAC pathway, standardised injury-specific patient information documentation, the development of an opt-in physiotherapy service, a standardised text message advising patients when their vFAC consultation will occur and the establishment of a referral pathway with a fracture liaison service. Future research is needed to explore how therapeutic relationships can be developed when care is delivered virtually and to explore the experiences of patient cohorts that were not included in this study.

## Supporting information

**S1 Checklist. Inclusivity in global research.**
(PDF)

**S2 Table. Purposive recruitment sample in regard to patients referred to the vFAC Pathway in 2023.**
(PDF)

**S3 Table. Interview guide.**
(PDF)

**S4 File. Participant recruitment process.**
(PDF)

## Acknowledgments

The authors wish to thank the PPI group for their support in developing the interview guide and the participants who gave up their time to complete this study.

## Author contributions

**Conceptualization:** Darragh Carolan, Shane O'Neill, Breon White, Gillian Yeowell.

**Data curation:** Darragh Carolan, Gillian Yeowell.

**Formal analysis:** Darragh Carolan, Gillian Yeowell.

**Investigation:** Darragh Carolan.

**Methodology:** Darragh Carolan, Shane O'Neill, Gillian Yeowell.

**Project administration:** Darragh Carolan, Shane O'Neill, Breon White, Frank Lyons, Grainne Colgan, Vinny Ramiah, Dervilla Danaher, Gillian Yeowell.

**Resources:** Darragh Carolan, Shane O'Neill, Breon White, Frank Lyons, Grainne Colgan, Vinny Ramiah, Dervilla Danaher.

**Supervision:** Shane O'Neill, Frank Lyons, Grainne Colgan, Vinny Ramiah, Dervilla Danaher, Gillian Yeowell.

**Validation:** Darragh Carolan, Shane O'Neill, Gillian Yeowell.

**Visualization:** Darragh Carolan, Shane O'Neill, Breon White, Frank Lyons, Grainne Colgan, Vinny Ramiah, Dervilla Danaher, Gillian Yeowell.

**Writing – original draft:** Darragh Carolan, Shane O'Neill, Breon White, Frank Lyons, Grainne Colgan, Vinny Ramiah, Dervilla Danaher, Gillian Yeowell.

**Writing – review & editing:** Darragh Carolan, Shane O'Neill, Breon White, Frank Lyons, Grainne Colgan, Vinny Ramiah, Dervilla Danaher, Gillian Yeowell.

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
