## [Decision Letter · Decision Letter 0]

16 Jan 2025

PONE-D-24-56296Patients’ experiences of a Virtual Fracture Assessment Clinic Pathway: A qualitative studyPLOS ONE

Dear Dr. Carolan,

Thank you for submitting this interesting and important work to PLOS One. Two researchers with clinical experience in virtual fracture clinics reviewed your paper and found it to be extremely clear and important. They only had minor comments for you to address. I also reviewed the paper and thought it was great. The background clearly set up the rationale for the study. The methods, results and conclusion clearly linked back to the aim, and the discussion brought up some very interesting and relevant points. My only comment is to improve the resolution of Fig 1 as it is quite blurry. Otherwise, well done on this important work!  We invite you to submit a revised version of the manuscript that addresses the points raised during the review process.

We look forward to receiving your revised manuscript.

Kind regards,

Joshua Robert Zadro, PhD

Academic Editor

PLOS ONE

Journal Requirements:

Reviewers' comments:

Reviewer's Responses to Questions

**Comments to the Author**

1. Is the manuscript technically sound, and do the data support the conclusions?

Reviewer #1: Yes

Reviewer #2: Yes

2. Has the statistical analysis been performed appropriately and rigorously? 

Reviewer #1: Yes

Reviewer #2: N/A

3. Have the authors made all data underlying the findings in their manuscript fully available?

Reviewer #1: Yes

Reviewer #2: No

4. Is the manuscript presented in an intelligible fashion and written in standard English?

Reviewer #1: Yes

Reviewer #2: Yes

5. Review Comments to the Author

Reviewer #1: To the authors,

This qualitative study provides insight to the experiences of patients with minor peripheral limb fractures utilizing the virtual fracture clinic pathway, from point of referral to post-discharge.

Well done! This research question is highly relevant and required. There is no similar studies of such in the current literature. The writing style is appropriate with minimal grammatical errors. The findings you had was compared with the current literature extensively.

These are some suggested minor grammatical changes:

1. P4 – Line 76: Add comma - In the vFRAC, an orthopaedic…

2. P4 – Line 77: Change ‘reviewed’ to ‘reviews’; change ‘determined’ to ‘determines’

3. P6 – Line 123: Add comma - …fracture clinic consultations, this…

4. P7 – Line 152: Change comma to full-stop – …vFAC pathway. This provided…

5. P15 – Line 322: Add comma - ….going on, it was just…

6. P15 – Line 323: Add comma - …a fracture, here you…

7. P16 – Line 341: Add comma & change word – it on a page, I’m ‘of’ that era…

8. P16 – Line 350: Add comma - …get it done, I like to…

9. P19 – Line 403: Add full stops - …take time off. I didn’t have…; …reschedule anything. I didn’t…

10. P19 – Line 418: Add full stops & comma - …call twice. I called…; …his call. I called him back, then we got…

11. P20 – Line 434: Please check the six overarching themes listed. There’s a duplicate of ‘reassurance’ and ‘severity of signs’ is missing.

12. P20 – Line 444: Change ‘is’ to ‘as’ – …may be a factor ‘as’ to why…

13. P25 – Line 539: Change ‘compliment’ to ‘complement’

14. P26 – Line 563: Change ‘this’ to ‘which’. Alternatively , change the comma to a full stop i.e. …the study. This may have…

Here are some other minor suggestions or questions to ponder:

1. Consider changing ‘peripheral fractures’ to ‘peripheral limb fractures’ for improved clarity.

2. In your Information section on page 15, you mentioned that participants highlighted the need for basic information on immobilsation devices. Did participants also mentioned the need for other information in the handouts such as their diagnosis, activity restrictions during the recovery period, and prognosis of recovery, or signs/symptoms to monitor e.g. deep vein thrombosis or complex regional pain syndrome?

3. Regarding fragility fractures on page 24, you could consider adding this information into the patient’s standardised injury/recovery handout so that they can initiate the conversation with their usual health practitioner on screening for osteoporosis?

4. In your results, you may want to add the mean (range) time of your interviews to provide readers a gauge on how in-depth the interviews were.

5. The addition of the S1 table is clear and reflects the closeness of your study participants’ demographics with the usual patients who utilise vFAC. Well done.

6. Thank you for sharing S3 Table – Distress Protocol. I’m not sure how important, or how this adds to this publication. You can consider removing this.

7. Finally, was there a general preference on which pathway patients preferred? I.e. did patients who were seen by a physiotherapist ‘happier’ than those who were ‘discharged’ vs those who were seen at the in-person clinic by a surgeon?

Overall, this is a well written paper, written by a team who has done extensive background reading, and presented the results very well. This will definitely fill the gaps in the literature to complement the vFAC research. I agree that it will be great to continue this research to better understand the experiences of patients from other backgrounds, or the carers of patients with cognitive impairments.

Reviewer #2: An interesting, well written paper addressing a research area which has been under explored. The paper explores the attitudes and concerns of patients experiences through the vFAC pathway and identifies areas of potential improvement.

The use of consumer engagement in developing the research topic and topic guide is a strength, especially as this research aims to have real world effects on patient care throughout the vFAC pathway. There is clear inclusion and exclusion criteria and the interviewed patients are similarly matched to the cohort of patients who received vFAC care in 2023, although there are interviews lacking from the 35-44yo cohort (0% vs 17%). I felt the finding of missed calls/unidentified phone calls to be particularly relevant due to the increasing scam calls and the increased awareness of these. Providing solutions to this issue could lead to practical changes to the delivery of care. Although it was only briefly discussed, I also thought the identification of the issues regarding ongoing referrals/identification of fragility fractures very relevant due to the increased awareness and importance of identifying these early. Further discussion may be warranted.

Weaknesses were as stated, the experiences of patients which were excluded and a few minor issues as stated below.

I am unable to view figure 1. (thematic map). I am not sure if this is a paper issue or user error.

Two minor spelling considerations below.

Line 341:

“I was quite happy with it on a page I’m off that era ((laughs)). Rita (Distal”

although this is a direct quote from a patient so possibly could have been what she said, should this instead have been transcribed as "I’m of that era"

Line 444:

“care is delivered virtually may be challenging and this may be a factor is to why”

be changed to "factor as to why"

6. PLOS authors have the option to publish the peer review history of their article (what does this mean?). If published, this will include your full peer review and any attached files.

Reviewer #1: **Yes: **Min Jiat Teng

Reviewer #2: **Yes: **Rowena Charteris

---

## [Author Response · Author response to Decision Letter 1]

1 Feb 2025

Response to points from academic editor:

Improve the resolution of Fig1:

The resolution of Fig1 has been improved using the recommended PACE digital diagnostic tool and uploaded to the editorial manager.

Please include a complete copy of PLOS’ questionnaire on inclusivity in global research in your revised manuscript:

Included (S1 Checklist. Inclusivity in Global Research Questionnaire Checklist.pdf).

We note that you have indicated that there are restrictions to data sharing for this study. For studies involving human research participant data or other sensitive data, we encourage authors to share de-identified or anonymized data. However, when data cannot be publicly shared for ethical reasons, we allow authors to make their data sets available upon request:

All relevant data are within the manuscript and supplementary information files. Relevant anonymised quotes from participants have been included in the results section of the manuscript. As per ethical approval obtained from the Mater Misericordiae University Hospital Institutional Review Board (reference number: 1/378/2387) and Manchester Metropolitan University Faculty Ethics Committee (reference number: 58689), anonymised versions of interview transcripts are saved on the Mater Misericordiae University Hospital computer system. Full anonymised transcripts can be made available on request. Requests can be sent to the Hannah King, Administrator, Institutional Review Board, Mater Misericordiae University Hospital, Eccles Street, Dublin 7, D07 R2WY, Ireland, hannahking@mater.ie, +353 18032971.

Response to points from reviewer #1:

All grammatical errors have been corrected as suggested (see revised manuscript with track changes).

1. Consider changing ‘peripheral fractures’ to ‘peripheral limb fractures’ for improved clarity:

Changed as suggested (see revised manuscript with track changes).

2. In your Information section on page 15, you mentioned that participants highlighted the need for basic information on immobilisation devices. Did participants also mentioned the need for other information in the handouts such as their diagnosis, activity restrictions during the recovery period, and prognosis of recovery, or signs/symptoms to monitor e.g. deep vein thrombosis or complex regional pain syndrome?

Participants only highlighted the need for more information about their diagnosis (due to receiving conflicting diagnoses at times) and using immobilisation devices, as mentioned in the results section. They did not mention a desire for information on diagnosis, activity restrictions, prognosis or signs/symptoms to monitor, perhaps as they felt this was adequately explained on the pathway.

3. Regarding fragility fractures on page 24, you could consider adding this information into the patient’s standardised injury/recovery handout so that they can initiate the conversation with their usual health practitioner on screening for osteoporosis?

This point has been added to the revised manuscript page 25, lines 535-537:

‘Including information on fragility fractures in standardised patient information leaflets may encourage patients to discuss bone health screening with their general practitioner.’

4. In your results, you may want to add the mean (range) time of your interviews to provide readers a gauge on how in-depth the interviews were.

Results section updated to include range of interview duration (30-75 minutes) and mean interview time (approximately 60 minutes). Page 12, lines 261-262.

5. The addition of the S1 table is clear and reflects the closeness of your study participants’ demographics with the usual patients who utilise vFAC. Well done.

No action required

6. Thank you for sharing S3 Table – Distress Protocol. I’m not sure how important, or how this adds to this publication. You can consider removing this.

Distress protocol removed from supplementary information.

7. Finally, was there a general preference on which pathway patients preferred? I.e. did patients who were seen by a physiotherapist ‘happier’ than those who were ‘discharged’ vs those who were seen at the in-person clinic by a surgeon?

There was no general preference on which pathway patients preferred. As mentioned in the results section, each participant had different preferences, i.e. some were happy to be referred to physiotherapy, while other were happy to self-manage their recovery. Similarly, some participants preferred to convenience of a virtual fracture clinic consultation, while others preferred attending an in-person fracture clinic.

Response to points from reviewer #2:

Interviewed patients are similarly matched to the cohort of patients who received vFAC care in 2023, although there are interviews lacking from the 35-44yo cohort (0% vs 17%):

This is a fair critique and a weakness in the sample recruited.

I am unable to view figure 1. (thematic map). I am not sure if this is a paper issue or user error.

The resolution of Fig1 has been improved using the recommended PACE digital diagnostic tool and uploaded to the editorial manager.

Two minor spelling considerations:

All spelling errors have been corrected as suggested (see revised manuscript with track changes).

---

## [Editor Report · Decision Letter 1]

5 Mar 2025

Patients’ experiences of a Virtual Fracture Assessment Clinic Pathway: A qualitative study

PONE-D-24-56296R1

Dear Dr. Carolan,

We’re pleased to inform you that your manuscript has been judged scientifically suitable for publication and will be formally accepted for publication once it meets all outstanding technical requirements.

Kind regards,

Joshua Robert Zadro, PhD

Academic Editor

PLOS ONE
---

## [Editor Report · Acceptance letter]

PONE-D-24-56296R1

PLOS ONE

Dear Dr. Carolan,

I'm pleased to inform you that your manuscript has been deemed suitable for publication in PLOS ONE. Congratulations! Your manuscript is now being handed over to our production team.

Kind regards,

on behalf of

Dr. Joshua Robert Zadro

Academic Editor

PLOS ONE